

# Tree variability limits the detection of nutrient treatment effects on sap flux density in a northern hardwood forest

Alexandrea M. Rice[1,2], Mariann T. Garrison-Johnston[3], Arianna J. Libenson[4] and Ruth D. Yanai[1]

[1] Sustainable Resources Management, SUNY College of Environmental Science and Forestry, Syracuse, NY, United States of America
[2] Geosciences, University of Massachusetts at Amherst, Amherst, MA, United States of America
[3] Ranger School, State University of New York College of Environmental Science and Forestry, Wanakena, NY, United States of America
[4] College of Arts and Sciences, University of Vermont, Burlington, VT, United States of America

Corresponding author
Ruth D. Yanai, rdyanai@syr.edu

## ABSTRACT

The influence of nutrient availability on transpiration is not well understood, in spite of the importance of transpiration to forest water budgets. Soil nutrients have the potential to affect tree water use through indirect effects on leaf area or stomatal conductance. For example, following addition of calcium silicate to a watershed at Hubbard Brook, in New Hampshire, streamflow was reduced for 3 years, which was attributed to a 25% increase in evapotranspiration associated with increased foliar production. The first objective of this study was to quantify the effect of nutrient availability on sap flux density in a nitrogen, phosphorus, and calcium addition experiment in New Hampshire in which tree diameter growth, foliar chemistry, and soil nutrient availability had responded to treatments. We measured sap flux density in American beech (*Fagus grandifolia,* Ehr.), red maple (*Acer rubrum* L.), sugar maple (*Acer saccharum* Marsh.), white birch (*Betula papyrifera* Marsh.), or yellow birch *(Betula alleghaniensis* Britton.) trees, over five years of experiments in five stands distributed across three sites. In 2018, 3 years after a calcium silicate addition, sap flux density averaged 36% higher in trees in the treatment than the control plot, but this effect was not very significant ($p = 0.07$). Our second objective was to determine whether this failure to detect effects with greater statistical confidence was due to small effect sizes or high variability among trees. We found that tree-to-tree variability was high, with coefficients of variation averaging 39% within treatment plots. Depending on the species and year of the study, the minimum difference in sap flux density detectable with our observed variability ranged from 46% to 352%, for a simple ANOVA. We analyzed other studies reported in the literature that compared tree water use among species or treatments and found detectable differences ranging from 16% to 78%. Future sap flux density studies could benefit from power analyses to guide sampling intensity. Including pretreatment data, in the case of manipulative studies, would also increase statistical power.

## INTRODUCTION

Transpiration is affected by vegetation type and coverage as well as environmental variables such as soil water availability and relative humidity (*Aber & Federer, 1992*; *Foley et al., 2000*; *Kite, 1998*). Deciduous and evergreen forest types differ in annual water use due to the length of the growing season (*Daley et al., 2007*) and tree species differ in sensitivity to drought stress (*Coble et al., 2017*; *Gu et al., 2015*). In addition, the availability of nitrogen (N), phosphorus (P), calcium (Ca), and silica may regulate transpiration rates (*Cramer, Hoffmann & Verboom, 2008*; *Ma, 2004*; *Matimati, Verboom & Cramer, 2014*), due to the role of these elements in stomatal opening (*Laanemets et al., 2013*; *Lautner et al., 2007*), leaf size (*Wilkinson, Bacon & Davies, 2007*), canopy reflectance (*Sullivan et al., 2013*) and root growth (*Wright et al., 2011*; *Wurzburger & Wright, 2015*).

Northern hardwood forests are poorly adapted to water stress due to the abundance of water in this region (*Pederson et al., 2014*). Models predict that the northeastern USA will experience more severe dry periods in the context of overall wetter conditions in the near future due to climate change (*Hayhoe et al., 2008*). Other studies have suggested that water limitation could become more important with intensifying summer droughts (*Brzostek et al., 2014*). Because of these projections it is increasingly important to understand the controls on water use in northern hardwood species. The effects of nutrient availability on water use are not currently represented in forest hydrology models because they are poorly understood.

There have been conflicting reports on the effects of nutrient availability on tree water use. An increase in sap flux density has been observed with the addition of multiple element fertilizers in Norway spruce in northern Sweden (*Phillips et al., 2001*) and in eucalyptus forests in Hawaii (*Hubbard et al., 2004*). Nitrogen addition reduced evapotranspiration rates in loblolly pine in North Carolina (*Ward et al., 2018*). Calcium additions (lime and gypsum) increased transpiration in central Amazonia (*Da Silva, Goncalves & Feldpausch, 2008*). Calcium silicate additions may have increased sap flow at Hubbard Brook in New Hampshire, based on an observation of decreased stream flow for 3 years, after which stream flow returned to pretreatment rates (*Green et al., 2013*).

In 13 stands distributed across three sites in the White Mountains of New Hampshire, N and P have been added in full factorial combination since 2011 to study Multiple Element Limitation in Northern Hardwood Ecosystems (MELNHE). By 2013, foliar nutrients had responded to treatment (*Wild & Yanai, 2015*). Relative basal area growth across all 13 stands responded to P addition but not to N addition by 2015, based on the average tree (*Goswami et al., 2018*), but the dominant trees grew more in response to N addition (*Hong et al., 2022*). The MELNHE study design included calcium silicate additions in seven of the stands to test the hypothesis that an increase in tree water use explained the reduction in runoff following the earlier calcium silicate addition at Hubbard Brook (*Green et al., 2013*).

One goal of this study was to determine the role of water use by trees in reducing runoff after a calcium silicate addition by measuring sap flux density of five common northern hardwood species in plots treated with calcium silicate at the same rate as the whole-watershed addition at Hubbard Brook (*Green et al., 2013*). We also investigated the

importance of other nutrients to tree water use by quantifying sap flux density in plots receiving additions of N and P. We predicted that sap flux density would increase with the addition of a limiting nutrient, because nutrient availability is a driver of productivity and photosynthesis. Additionally, we calculated minimum detectable differences in this study and in previously published studies to determine whether failures to detect treatment effects were due to small effect sizes or to high variability among trees. Power analyses of studies such as these can help to guide future research plans.

## METHODS

### Site description

We studied tree water use in five naturally regenerated hardwood stands located in three forested sites in the White Mountain National Forest, New Hampshire, USA: two in each of the Bartlett Experimental Forest (44°02′N, 71°17′W) and Hubbard Brook Experimental Forest (43°93′N, 71°73′W), and one in Jeffers Brook (44°03′N, 71°88′W; Table 1). The climate is humid continental, with a mean annual temperature of 4.4 °C and precipitation of 1400 mm (*Bailey et al., 2003*; *Smith & Martin, 2001*). Soils are moderately well to well drained (*Schaetzl et al., 2009*), coarse-loamy Spodosols and Inceptisols developed in glacial drift derived from granitic and metamorphic silicate rocks (*Vadeboncoeur et al., 2012*; *Vadeboncoeur et al., 2014*). Dominant tree species are American beech (*Fagus grandifolia* Ehrh.), sugar maple (*Acer saccharum* Marsh.), and yellow birch *(Betula alleghaniensis* Britton) in mature stands with the inclusion of white birch (*B. papyrifera* Marsh.) and red maple (*A. rubrum* L.) in successional stands (*Fatemi et al., 2012*; *Naples & Fisk, 2010*). Tree height ranged from 9.9 m to 21.3 m in the successional stands and 10.3 m to 33.2 m in the mature stands.

These five stands are part of a study of Multiple Element Limitation in Northern Hardwood Ecosystems (Table 1; *Hong et al., 2021*). Within each of these stands there are four plots measuring 50 × 50 m, except for Hubbard Brook Successional, which has 30 × 30 m plots. These plots have been treated since 2011 with 30 kg ha$^{-1}$yr$^{-1}$ of N as $NH_4NO_3$, 10 kg ha$^{-1}$yr$^{-1}$ of phosphorus as $NaH_2PO_3$, both N and P, or were left untreated. A fifth plot in these stands received a one-time application of 1,150 ha$^{-1}$ of calcium silicate (wollastonite) in 2011, or in 2015 for Hubbard Brook Successional.

### Field methods

Sap flux density measurements were taken during the summers of 2013, 2014, 2015, 2017, and 2018 for periods of 7 to 48 days. In 2013 and 2014, we examined three tree species in three mature stands in plots that received calcium silicate and control treatments (six plots total; Table 2). In 2015, our goal was to assess the effects of N and P treatments (four plots) on one species in one successional stand (Table 3). During those years, data were sparse due to poor contact between the probes and the sapwood, but this problem was corrected in 2017. In 2017, one species in a mature stand in Bartlett was examined, and in 2018, three species were examined in the Hubbard Brook Successional stand. The number of trees of each species monitored in each plot varied by stand and year (Tables 2 and 3).

**Table 1  Site descriptions of the five forested stands studied in the White Mountains of New Hampshire.** Date of stand initiation was determined from the last clearcut or heavy harvest event. Quadratic mean diameters and basal areas are based on trees with diameters greater than 2 cm. Stem density is calculated for trees greater than 10 cm in diameter.

| Stand | Date of stand initiation | Elevation (m) | Slope (%) | Aspect | Quadratic mean diameter (cm) | Stem density (count/ha) | Basal area (m²/ha) |
|---|---|---|---|---|---|---|---|
| Bartlett Successional (C6) | 1975 | 460 | 13–20 | NNW | 23.1 | 1,487 | 29.5 |
| Bartlett Mature (C8) | ~1883 | 330 | 5–35 | NE | 41.2 | 589 | 41.0 |
| Jeffers Brook Mature | ~1900 | 730 | 30–40 | WNW | 35.4 | 680 | 35.9 |
| Hubbard Brook Successional | 1971 | 500 | 10–25 | S | 23.6 | 2749 | 15.4 |
| Hubbard Brook Mature | ~1910 | 500 | 25–35 | S | 36.5 | 538 | 29.5 |

**Table 2  Studies of sap flux density in response to calcium silicate addition showing the dates, number of years since treatment and the number of trees of each species used in analysis in each treatment plot.** The calcium silicate treatment was applied in 2011, except that in Hubbard Brook Successional it was applied in 2015.

| Dates (# of years post treatment) | Stand | Species | Number of trees in treatment | |
|---|---|---|---|---|
| | | | Control | Calcium Silicate |
| July 19, 2013 (2) | Jeffers Brook Mature | Sugar maple | 4 | 2 |
| | | Yellow birch | 3 | 1 |
| August 5, 2013 (2) | Bartlett Mature (C8) | American beech | 2 | 3 |
| | | Sugar maple | 3 | 3 |
| August 5, 2013 (2) | Hubbard Brook Mature | American beech | 2 | 2 |
| | | Sugar maple | 2 | 2 |
| | | Yellow birch | 2 | 1 |
| June 22–24, 2014 (3) | Bartlett Mature (C8) | Sugar maple | 3 | 2 |
| July 1, 2014 (3) | Hubbard Brook Mature | American beech | 3 | 2 |
| | | Sugar maple | 3 | 2 |
| | | Yellow birch | 3 | 1 |
| August 1, 2 and 5, 2014 (3) | Jeffers Brook Mature | Sugar maple | 4 | 4 |
| | | Yellow birch | 3 | 5 |
| July 22 and 23, 2015 (4) | Bartlett Successional (C6) | White birch | 4 | 4 |
| July 31 and August 1, 2017 (6) | Bartlett Mature (C8) | Sugar maple | 5 | 4 |
| July 19–21, 2018 (3) | Hubbard Brook Successional | Red maple | 3 | 3 |
| | | Sugar maple | 3 | 3 |
| | | White birch | 3 | 3 |

Sap flux density measurements were made using the thermal dissipation method (*Granier, 1987*). This technique used a pair of stainless-steel probes 20 mm long and 1.8 mm in diameter. One probe was wrapped with copper constantan thermocouple wire (Type T) and generated a constant flow of heat (0.2 W). The other was a reference probe that received no heat. Both probes were coated with heat-conducting paste and inserted into aluminum sleeves 20 mm long and 2.4 mm in diameter for protection during insertion into the tree. The thermal dissipation method may underestimate sap flux density depending on species, portion of probe in the heartwood, and wood type (*Flo et al., 2019*; *Peters et al., 2018*), but a consistent bias would not impair our ability to detect treatment effects. We

**Table 3** Studies of sap flux density in response to nitrogen and phosphorus addition showing the date of the study, the number of annual nutrient applications prior to each study, and the number of trees of each species used in analysis in each treatment plot.

| Dates (# of years of treatment) | Stand | Species | Number of trees in treatment | | | |
|---|---|---|---|---|---|---|
| | | | Control | Nitrogen | Phosphorus | Nitrogen + Phosphorus |
| August 1, 2015 (4) | Bartlett Successional (C6) | White birch | 4 | 4 | 5 | 4 |
| July 31 and August 1, 2017 (6) | Bartlett Mature (C8) | Sugar maple | 5 | 6 | 6 | 5 |
| July 19–21, 2018 (7) | Hubbard Brook Successional | Red maple | 3 | 3 | 3 | 3 |
| | | Sugar maple | 3 | 3 | 3 | 3 |
| | | White birch | 3 | 3 | 3 | 3 |

tested for treatment differences in sap flux in the youngest xylem; reporting sap flow per tree or scaling to the stand level would require information on sapwood depth, which was not measured.

We selected trees with full canopies, without noticeable dead branches or cankers. A total of 202 trees were instrumented, of which 174 produced usable data. The number of trees instrumented per species per plot ranged from 1 to 6, averaging $3.2 \pm 1.1$ (standard deviation; Tables 2 and 3). Before probe installation, $\sim 4$ cm$^2$ of bark was removed to the cambium at a height of 1.37 m for the reference probe and 1.47 m for the heated probe on the south-facing side of the tree. After the bark was removed, a hole 21 mm deep and 2.8 mm in diameter was drilled in the middle of the exposed cambium for each probe.

Once probes were inserted, they were protected from precipitation with a plastic cover sealed with acid-free silicone caulk. Closed-cell reflective polyethylene insulation was stapled over the plastic cover to shield solar radiation. The probes were connected to cables no longer than 20 m to minimize loss of power from cable resistance, trampling by humans, and disturbance by wildlife.

Temperature differences between the heated and reference probes in each plot were recorded every 15 min as an average of thirty 30-second readings on a multiplexor and stored on a multichannel data logger (Campbell Scientific CR800). To power the data logger and multiplexer, the first study, in 2013, used two 12 V deep-cycle marine batteries per plot charged by two solar panels; later studies used three batteries linked together.

Gaps in usable data were usually due to cable disturbances or improper probe installation; in the first year solar panels proved insufficient to maintain their charge, and data were very spotty. The number of days of usable data was also improved by more frequent maintenance of wire connections, probes, and batteries.

### Data processing

Temperature differences were converted into sap flux density using BaseLiner (version 3.0.10, developed by Ram Oren, Duke University; *Oishi, Hawthorne & Oren, 2016*). BaseLiner used the following equation to calculate sap flux density (g H$_2$O m$^{-2}$ of sapwood day$^{-1}$):

$$\text{Sap Flux Density} = 119 \times \left( \frac{\Delta T\,\text{max} - \Delta T}{\Delta T} \right)^{1.23} \tag{1}$$

where the constants, 119 (Watts $°C^{-1}$) and 1.23, were derived using the quantity of heat applied to the probes (*Granier, 1985*; *Lu, Urban & Ping, 2004*). $\Delta T_{max}$ (°C) is the "baseline" value that was manually chosen as the maximum temperature difference between the two probes from 10:00 PM to 4:30 AM EDT. $\Delta T_{max}$ was set for each day for each tree, defining nocturnal flux as zero, which accounted for any thermal drift that may have occurred from day to day. $\Delta T$ (°C) is the temperature difference between the probes at each 15-minute observation period. We used the same equation for all trees because species-specific coefficients were available for only one of the five species studied (*Peters et al., 2021*).

The experimental unit was a treatment plot, and the observational unit was a tree. Because of the many barriers to continuous collection of high-quality data, selecting data for analysis was an important step. Twenty-eight trees (not shown in Tables 2 and 3) never produced usable data due to faulty probe installation in the early years of the study. For each tree, we used the average daily sap flux density calculated from summed 15-minute averages for each day. Days used in analyses were mostly dry and sunny (PAR > 800 W/m$^2$ for at least 7 hours). Days were excluded from analyses if sap flux values did not follow the characteristic diurnal curve or if the majority of the probes in the stand were not functioning simultaneously (Tables 2 and 3).

## Data analysis

The five studies of calcium silicate addition were analyzed in ANOVA Type III sum of squares with average daily sap flux density as the response variable and treatment, species, and the number of years post treatment as categorical fixed effects. The post-treatment years were nested within two categories: "early" and "late", based on the finding that evapotranspiration increased for 3 years after a calcium silicate addition and then returned to normal (*Green et al., 2013*). Treatments were nested within stands and stands were treated as a random effect. The assumption of normality of residuals was met through a logarithmic transformation of sap flux density. We tested a model that included additional interaction terms, but the Akaike's Information Criterion (AIC), 44.4, was higher than that of the simpler model (1.48) (Table S1). Data from 2018 were analyzed independently using the same model without time since treatment. For this analysis, the assumption of normality of residuals was met without transformation.

Our study design included all combinations of N and P addition, but no combinations of calcium with N or P. To take advantage of the factorial N × P addition, we analyzed the N and P treatments separately from the calcium silicate addition treatments. Thus, the three studies in N and P addition plots were analyzed together in a randomized full factorial ANOVA with average daily sap flux density as the response variable and species and stand as fixed effects. Three stands were studied in three different years, which precluded distinguishing the effects of stands from the effects of the year of measurement. A model including the interaction of species and treatment was not an improvement, according to the AIC (97 compared to 86) (Table S2). Normality of the residuals was achieved by logarithmic transformation of sap flux density. Additionally, the year with the best data quality (2018) was analyzed independently using the same model with the exception of excluding stand as a fixed effect. For this analysis, the assumption of normality of residuals

was met without transformation. We tested whether nutrient additions (Ca and N × P) affected sap flux density at any time of day, using trees from 2017, which had the largest sample size of a single species (sugar maple). These effects were tested for each hour over three days using ratios of instantaneous sap flux density to the average for the day. There were no significant treatment effects when a Bonferroni correction was applied ($p \geq 0.32$). Therefore, only results using daily average sap flux densities were reported.

All analyses were conducted using the "lme4" package in *R* version 3.5.2 (*R Core Team, 2018*).

Our second objective was approached by calculating coefficients of variation (CV) and minimum detectable differences for our study and published studies. Coefficients of variation across trees were calculated for each treatment by stand and year studied using the average daily sap flux density per tree. Coefficients of variation across treatments were also calculated for each stand and year studied for comparison to within-treatment CV's. Published studies were chosen for minimum detectable difference analyses from a search using the keywords "sap flux", "sap flow", "trees", and "treatment addition", and were selected if they reported sample sizes, mean sap flux density, and standard deviation or error for each treatment or species of interest.

The minimum detectable difference in sap flux density among treatments or species in our study and for published studies was calculated using the following equation (*Zar, 1984*):

$$\text{Minimum Detectable Difference} = \sqrt{\frac{2kS^2\phi^2}{n}} \qquad (2)$$

where $k$ is the number of treatments, $S^2$ is the sample variance, $\phi^2$ is the non-centrality parameter dependent on $\beta$ and $\alpha$, and n is the sample size. Power (1- $\beta$) was set at 0.8 and $\alpha$ at 0.05. For an ANOVA, $S^2$ is the mean squared error. This equation was rearranged to calculate the sample size needed to detect a 20% and 50% difference using data from 2017 and 2018. To estimate the minimum detectable difference from published studies, $S^2$ was the average of the variances across treatments (Eq. (2)).

## RESULTS

### Characteristics of sap flux density

Sap flux density peaked between 12:30 and 3:00 PM and was lowest between 2:30 and 4:30 AM Eastern Daylight Savings Time. Trees varied consistently in sap flux density across days (Fig. 1).

The first year, 2013, had the highest CV's among the five studies (Table 4). Coefficients of variation across trees within plots for each year ranged from 9% to 136% and averaged 38% within treatment plots across trees. CV's across treatments averaged 17% with a range of 8% in Bartlett Mature in 2017 to 49% in Jeffers Brook Mature in 2013. Average CV's within treatment plots were higher than CV's across treatments.

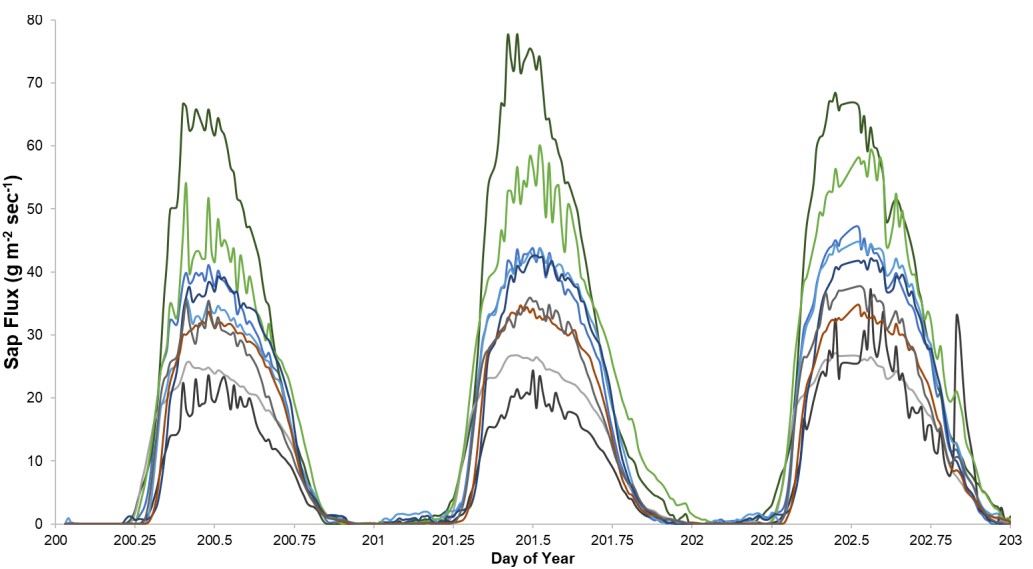

**Figure 1** An example of sap flux density in nine trees continuously measured from July 19 to 21, 2018. Each color represents one tree in the control plot in the Hubbard Brook Successional stand.

## Effects of nutrient additions

In 2018, the year with the best data quality, the effect of Ca addition was a marginally significant increase of 36% ($p = 0.07$). With all years included, the effect of calcium silicate addition on sap flux density was not consistent ($p = 0.30$ for the main effect of treatment). Sap flux density did not differ consistently between "early" (2- and 3-years post treatment) and "late" (4 and 6 years post treatment) years ($p = 0.26$) nor was there a difference in response to a calcium silicate addition as the number of years post treatment increased ($p = 0.88$ for the interaction of treatment and time period). The five species were indistinguishable in sap flux density ($p = 0.42$; Fig. 2; Table 5).

The effects of N and P on sap flux density were studied in white birch in Bartlett Successional in 2015, sugar maple in Bartlett Mature in 2017, and red maple, sugar maple, and white birch in Hubbard Brook Successional in 2018 (Table 3). Sap flux density differed across these three studies ($p < 0.01$), which could be because sites were different or because conditions differed during the measurement periods among the three years. Sap flux density in the Hubbard Brook Successional stand, measured in 2018 after 7 years of fertilization, was 63% higher than in Bartlett Successional measured in 2015 and Bartlett Mature measured in 2017. There was, however, no detectable difference in sap flux density due to the main effects of N ($p = 0.50$) or P ($p = 0.95$) or their interaction ($p = 0.33$). Species were also indistinguishable in sap flux density ($p = 0.58$; Figs. 3 and 4; Table 6).

In 2018, the best year for data quality, there was still not a consistent effect of N ($p = 0.26$) or P ($p = 0.10$) addition on sap flux density. Trees in the N and P plots had 14 and 20% higher sap flux densities, respectively, than those in the controls but trees in the NP plot did not, resulting in a weak N × P treatment interaction ($p = 0.06$). This is apparent in Figs. 3 and 4, where the filled blue symbols, showing the plots receiving the element not

**Table 4 Coefficient of variation (CV) of sap flux for each treatment plot in each study year.** CV's were calculated as the standard deviation of sap flux for all the trees in a treatment plot divided by the average for that plot.

| Year | Stand | Treatment | CV (%) |
|------|-------|-----------|--------|
| **2013** | Bartlett Mature (C8) | Control | 136 |
| | | Calcium | 58 |
| | Hubbard Brook Mature | Control | 36 |
| | | Calcium | 32 |
| | Jeffers Brook Mature | Control | 34 |
| | | Calcium | 32 |
| **2014** | Bartlett Mature (C8) | Control | 36 |
| | | Calcium | 34 |
| | Hubbard Brook Mature | Control | 40 |
| | | Calcium | 45 |
| | Jeffers Brook Mature | Control | 16 |
| | | Calcium | 35 |
| **2015** | Bartlett Successional (C6) | Control | 14 |
| | | Nitrogen | 47 |
| | | Phosphorus | 30 |
| | | Nitrogen + Phosphorus | 14 |
| | | Calcium | 9 |
| **2017** | Bartlett Mature (C8) | Control | 31 |
| | | Nitrogen | 29 |
| | | Phosphorus | 21 |
| | | Nitrogen + Phosphorus | 67 |
| | | Calcium | 44 |
| **2018** | Hubbard Brook Successional | Control | 30 |
| | | Nitrogen | 36 |
| | | Phosphorus | 47 |
| | | Nitrogen + Phosphorus | 37 |
| | | Calcium | 31 |

depicted on the $x$ and $y$ axes, fall below the open symbols. However, the interaction is based on only one plot receiving both treatments and is thus not very convincing: in this case, an environmental factor that differentially affected water use in one treatment plot would explain the observed "interaction" of N and P, whereby the NP plot had lower values than predicted by the main effects of N and P, which were positive, if not significant. The $p$ value of 0.10 for the main effect of P seems possibly worthy of attention, but both of the two main effects became less significant when the interaction term was excluded from the model ($p = 0.66$ for P), again showing that the analysis based on only one stand is not very robust.

## Power analyses

The lack of significant treatment effects does not necessarily mean that effect sizes were small (*Amrhein, Greenland & McShane, 2019*); rather, the minimum detectable differences
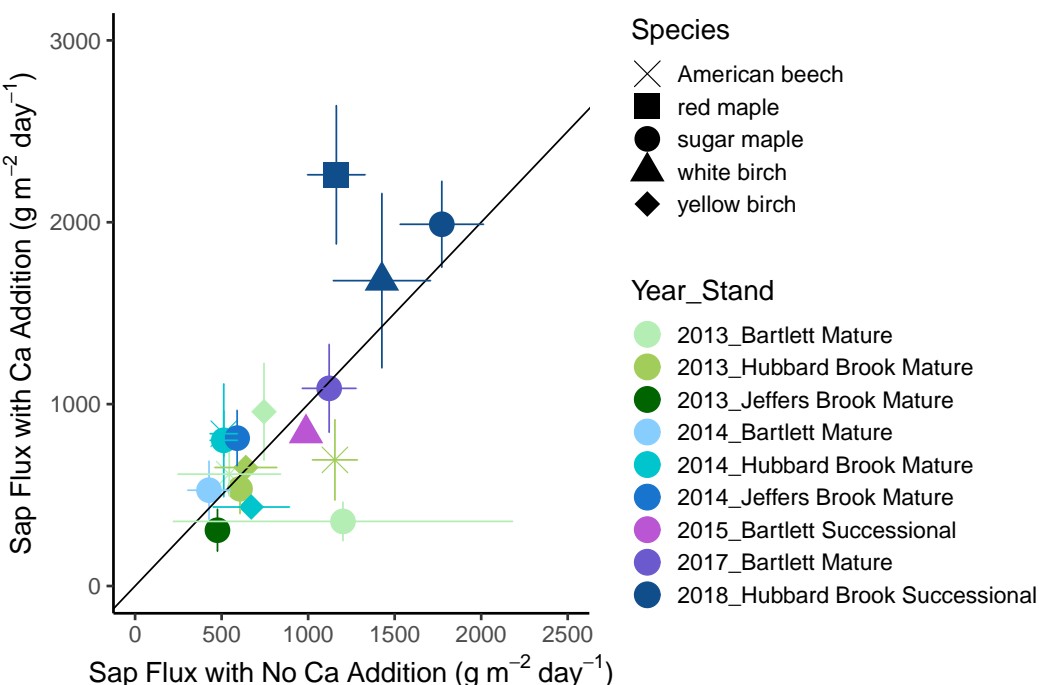

**Figure 2 Response of sap flux to calcium silicate treatment.** Bars represent standard errors across trees. Symbols represent species and colors represent years with associated stands. The 1:1 line represents equal flow rates in control and Ca-treated plots.

**Table 5 ANOVA table showing Type III sum of squares for calcium silicate addition studies.** The response variable is sap flux density, with calcium silicate treatment, species, and "Early" vs "Late" as categorical fixed effects. "Early" measurements were taken 2 or 3 years after the calcium silicate addition and "Late" were taken 4 or 6 years after the addition (Table 2).

|  | Df Numerator | Df Denominator | F Value | Pr (>F) |
|---|---|---|---|---|
| "Early" vs "Late" | 1 | 333.77 | 1.2932 | 0.26 |
| Treatment | 1 | 6.14 | 1.2764 | 0.30 |
| "Early" vs "Late × Treatment | 1 | 8.45 | 0.0227 | 0.88 |
| Species | 4 | 20.27 | 1.0287 | 0.42 |

for a simple ANOVA were large, ranging from 50 to 1582 g m$^{-2}$ day$^{-1}$ (50% to 352% of the mean) for Ca additions and from 779 to 1209 g m$^{-2}$ day$^{-1}$ (46% to 134%) for N × P additions, depending on the year (Fig. 5). Improvements in methods and larger sample sizes resulted in smaller detectable differences over time, associated with lower variability from tree to tree (Table 4). Even in 2018, where detectable differences were smallest, the variation from tree to tree within a plot was large (Fig. 6).

The average number of trees monitored per plot increased over time from two in 2013 to nine in 2018. Even with this increase in sample size, we were still unable to detect a treatment effect. We calculated the number of trees needed to detect a given effect size, using the studies with the best data quality, and assuming there were no differences among

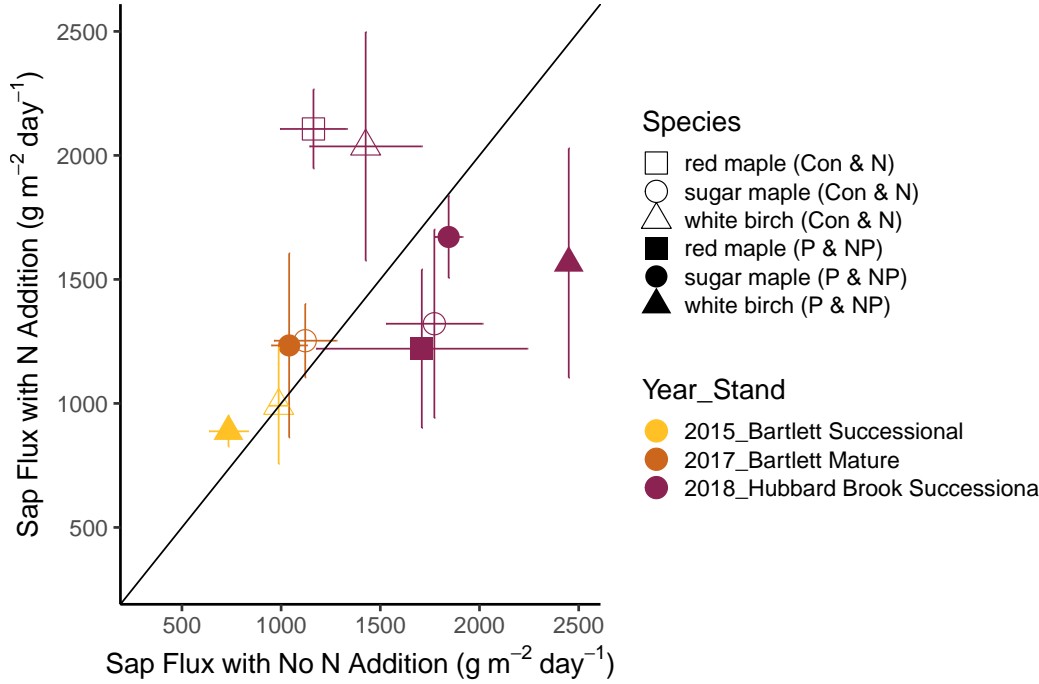

**Figure 3** **Responses of sap flux to nitrogen treatments.** Bars represent the standard error across trees. Species are represented by symbols; stands with the year studied are represented by colors. The 1:1 line represents equal sap flux rates in plots that received nitrogen additions (N and NP) and plots without nitrogen additions (Control and P).

species. To detect a 20% treatment effect on sap flux with 95% confidence would have required 119 trees in each treatment plot in 2017 or 48 trees in each plot in 2018. The number of trees needed to detect a 50% difference would be 19 for 2017 and 9 for 2018, given the variance in sap flux density we measured in those years.

## Minimum detectable differences for published studies

Effect sizes reported in eight previously published studies of tree water use ranged from 8% to 243% (Table 7). The largest reported effect size was a 90% increase in sap flux density with the addition of N ($80 \, \text{kg ha}^{-1} \, \text{yr}^{-1}$) in loblolly pine (*Pinus taeda*) seedlings (*Samuelson et al., 2008*). A 43% increase was reported with the addition of macro and micronutrients to a *Eucalyptus saligna* plantation in Hawaii (*Hubbard et al., 2004*), and a 35% increase was detected in a study involving the addition of P ($50 \, \text{kg ha}^{-1}$ of $P_2O_5^{-1}$) and Ca (2 t ha$^{-1}$ of $CaCO_3$) on *Vismia japurensis, Bellucia grossularioides*, and *Laetia procera* in Amazonia (*Da Silva, Goncalves & Feldpausch, 2008*).

Minimum detectable differences ranged from 16% to 75% in temperate forests and 52% to 74% in tropical and subtropical forests (Table 7). Two studies of temperate deciduous forest species had lower calculated detectable differences than ours. A study of eight tree species in four forest types in Wisconsin had a detectable difference in sap flux density of only 16% with a sample size of 8 per species, due to low variability among trees (*Ewers et al., 2002*). Another study in the same region used more trees from a larger area, resulting

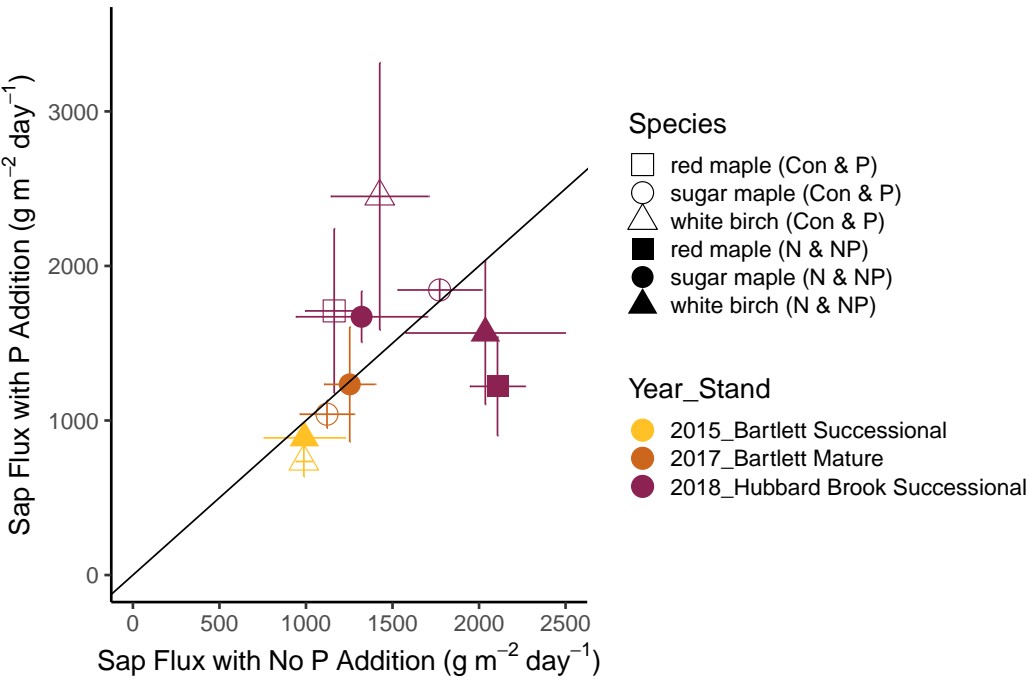

**Figure 4** Responses of sap flux to phosphorus treatments. Bars represent the standard error across trees. Species are represented by symbols; stands with the year studied are represented by colors. The 1:1 line represents equal sap flux rates in plots that received phosphorus additions (P and NP) and plots without phosphorus additions (Control and N).

**Table 6** ANOVA table showing Type III sum of squares for N and P addition studies. The response variable is average daily sap flux density and the listed explanatory variables are categorical fixed effects. The three stands were studied in three different years (Table 3).

|  | Sum of squares | Df | *F* Value | Pr (>*F*) |
|---|---|---|---|---|
| N Treatment | 0.071 | 1 | 0.460 | 0.50 |
| P Treatment | 0.001 | 1 | 0.004 | 0.95 |
| N × P Interaction | 0.146 | 1 | 0.944 | 0.33 |
| Species | 0.168 | 2 | 0.543 | 0.58 |
| Stand | 4.481 | 2 | 14.478 | <0.001 |
| Error | 10.987 | 71 | | |

in higher variability and slightly higher detectable differences (*Loranty et al., 2008*). Some studies had much higher effect sizes than ours (*Kunert, Schwendenmann & Hölscher, 2010*; *Nagler, Glenn & Thompson, 2003*; *Samuelson et al., 2008*) and thus were able to detect them with statistical confidence although their power to detect a difference was not better. Publication of insignificant findings is rare (*Møller & Jennions, 2001*); one earlier study took place in two MELNHE stands (including one that we studied, Bartlett Mature) with effect sizes well below detection, similar to ours (*Hernandez-Hernandez, 2014*).

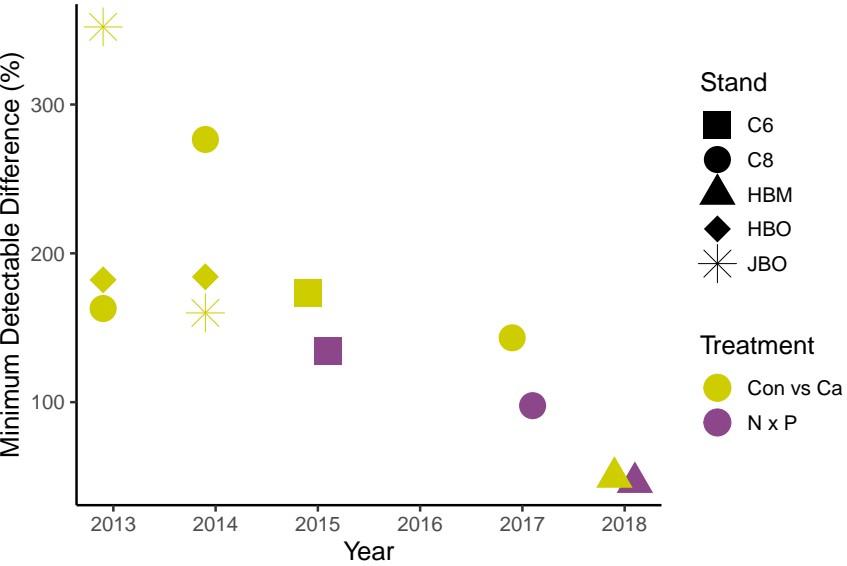

**Figure 5  Minimum detectable treatment differences for each stand, based on a simple ANOVA.** Shapes represent stands and colors represent treatment comparisons. Control and calcium plots were studied from 2013 to 2018. Nitrogen and phosphorus additions were studied from 2015 to 2018. Detectable differences for the factorial N × P ANOVA are somewhat smaller than shown here.

# DISCUSSION

The initial impetus for this study was the surprising decrease in stream flow for three years following a calcium silicate addition at Hubbard Brook, where tree water use was invoked as a likely mechanism (*Green et al., 2013*). Unfortunately, even in 2018, the year with the lowest tree-to-tree variability and the highest replication, our statistical power was too low to detect a response of sap flux density to calcium addition smaller than 50% with an $\alpha$ of 0.05 (Fig. 5); we obtained a $p$ value of 0.07 for an average increase in sap flux density of 36%, which is larger than the 25% increase in water use postulated by *Green et al. (2013)*. Thus, we lack the statistical confidence to either support or contradict the explanation that increased sap flux density accounted for the decline in runoff in the earlier calcium addition experiment.

There are other possible mechanisms for increased water use in response to nutrient addition. In the MELNHE study, the relative basal area increment of the average tree increased in response to P addition (*Goswami et al., 2018*), and larger trees grew more in response to N addition (*Hong et al., 2022*). In the calcium silicate addition, trees grew more than in the unfertilized control (*Battles et al., 2014*; *Huggett et al., 2007*). An increase in diameter growth could result in increased sapwood area (*Nilsson et al., 2021*), which could conduct water to a greater leaf area (*Wang et al., 2012*) without an increase in sap flux density (*Hubbard et al., 2004*). It is also possible that nutrient availability affects the length of the growing season (*Escudero et al., 1992*), which would affect annual water use without affecting sap flux density.

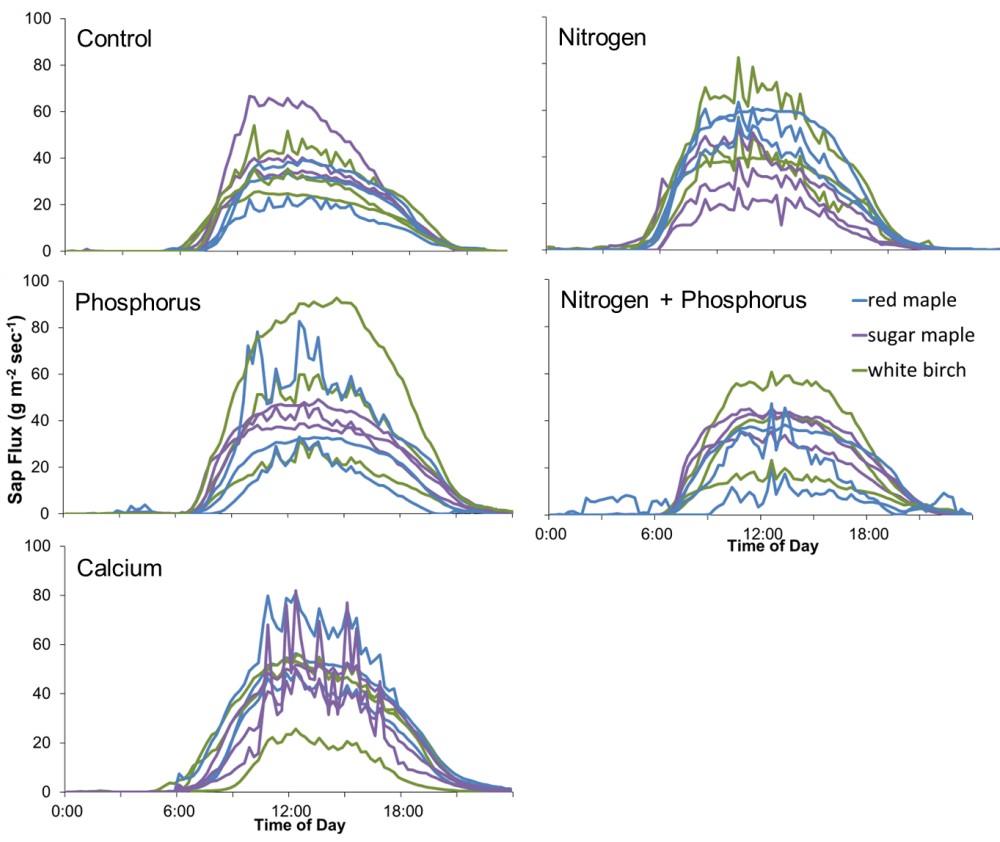

**Figure 6** **Diurnal sap flux density in the Hubbard Brook Successional stand on July 19, 2018.** Times are local daylight savings time. Each line represents an instrumented tree; there are 9 trees in each treatment colored by species.

Other studies of sap flux density have reported statistically significant effects to treatment additions, with effect sizes as low as 35% (Table 7). The difference detectable with our factorial ANOVA comparing N and P additions is somewhat smaller than the 46% shown in Fig. 5, because Eq. (1) describes power for a simple ANOVA. The benefit of factorial designs is evidenced by two of the published studies we analyzed. The study involving combinations of phosphate, lime, and gypsum additions (*Da Silva, Goncalves & Feldpausch, 2008*) was able to detect a significant difference of 35%, whereas our calculated minimum detectable difference for a simple ANOVA was 52%. Similarly, the full factorial study of irrigation and N fertilization (*Samuelson et al., 2008*) reported a 90% increase with N fertilization with a $p$-value of 0.01, which shows greater power than the detectable difference of 97% calculated using Eq. (2) for a simple ANOVA with $\alpha = 0.05$. Unfortunately, without a power analysis for factorial designs, we cannot better quantify the effect size ruled out by our findings in the N × P addition experiment. It remains possible that nutrient availability has an ecologically important effect on sap flux.

Our study might have benefitted from pretreatment measurements since sap flux density varied consistently from tree to tree (Figs. 1 and 6). Pretreatment data, like factorial designs,

**Table 7** Studies of sap flux density or sap flow among species or following nutrient additions in deciduous and evergreen trees, listed from lowest to highest minimum detectable difference.

| Forest type | Species or treatment studied (number of trees) | Effect size and significance | Minimum detectable difference (%) | Source |
|---|---|---|---|---|
| | | *Sap Flux Density* | | |
| Temperate Deciduous and Evergreen | *Pinus resinosa* (8) *Pinus banksiana* (8) *Acer saccharum* (8) *Populus tremuloides* (8) *Abies balsamea* (8) *Thuja occidentalis* (8) *Abies balsamea* (8) *Alnus regosa* (8) | Species was significant overall; pairwise comparisons ranged from 7.5 to 243% ($p < 0.05$) | 16 | *Ewers et al. (2002)* |
| Temperate Deciduous and Evergreen | *Alnus incana* (41) *Populus tremuloides* (79) *Thuja occidentalis* (9) | Not Reported | 21–65 | *Loranty et al. (2008)* |
| Temperate Evergreen (*Eucalyptus saligna*) | Control (9) Micro and macronutrients (9) | 43% increase with fertilization ($p = 0.04$) | 40 | *Hubbard et al. (2004)* |
| Temperate Deciduous | Control (9) Nitrogen (9) Phosphorus (9) Nitrogen + phosphorus (9) Calcium (9) | 14% increase with N ($p = 0.26$); 20% increase with P (0.10) 36% increase with Ca treatment ($p = 0.07$) | 46 (N × P) 50 (Ca) | This study (2018 only) |
| Tropical Evergreen (*Vismia japurensis, Bellucia grossularioides, Laetia procera*) | Control (9) phosphate (9) phosphate + lime (9) phosphate + lime + gypsum (9) | Phosphate + lime + gypsum treatment was 35% higher than phosphate + lime ($p < 0.05$) | 52 | *Da Silva, Goncalves & Feldpausch (2008)* |
| Temperate Deciduous | *Quercus mongolica* (5) *Tilia amurensis* (5) *Ulmus davidiana* (5) *Cornus controversa* (3) *Acer mono* (3) | No Significance ($p = 0.73$) | 52–75 | *Jung et al. (2011)* |
| Tropical Deciduous | *Cedrela odorata* (4) *Anacardium excelsum* (4) *Hura crepitans* (4) *Luehea seemannii* (4) *Tabebuia rosea* (4) *Gmelina arborea* (4) *Tectona grandis* (4) *Acacia mangium* (4) *Terminalia amazonia* (4) | *A. excelsoum, L. seemannii* and *T. amazonia* were 113% higher than *C. odorata* and *G. arborea* ($p < 0.05$) | 61 | *Kunert, Schwendenmann & Hölscher (2010)* |
| Temperate Deciduous | Control 7) Nitrogen (9) Phosphorus (9) Nitrogen + phosphorus (9) | 12% increase with N and 12% increase with P ($p > 0.05$) | 66 | *Hernandez-Hernandez (2014)* |

**Table 7** (*continued*)

| Forest type | Species or treatment studied (number of trees) | Effect size and significance | Minimum detectable difference (%) | Source |
|---|---|---|---|---|
| | | *Sap Flow Rate* | | |
| Subtropical Evergreen (*Pinus taeda*) | Control (5) Irrigated (5) Nitrogen (5) Irrigated + Nitrogen (5) | 90% increase with N fertilization ($p < 0.01$) | 74 | *Samuelson et al. (2008)* |
| Riparian Deciduous (Desert) | *Populus fremontii* (6) *Salix gooddingii* (6) *Tamarix ramosissima* (6) | *T. ramosissima* was 122% higher than *P. fremontii* and *S. gooddingii* ($p < 0.05$) | 78 | *Nagler, Glenn & Thompson (2003)* |

can confer additional power. The only study we analyzed that collected pretreatment data (*Hubbard et al., 2004*) reported an effect size of 43% with $p = 0.04$, which seems consistent with the minimum detectable difference we calculated (40% with $\alpha = 0.05$). Thus, it seems unlikely that the pretreatment data were used in the statistical analysis.

In addition to including pre-treatment measurements, future sap flux density studies could benefit from incorporating explanatory variables such as leaf area, canopy position, and stomatal conductance, since these characteristics have important influences on sap flux density (*Green et al., 2013*; *Hubbard et al., 2004*; *Peters et al., 2021*; *Phillips et al., 2001*). Environmental variables such as soil moisture and light environment may also be of interest since they influence sap flux density (*Oren et al., 1998a*; *Oren et al., 1998b*). Sapwood area measurements not only allow for scaling up to whole-tree water use but also make it possible to ensure that probes are within the sapwood. If probes are inserted into heartwood, which may have occurred in our study, estimated sap flux measurements can be reduced up to 50% (*Clearwater et al., 1999*). Many of these variables likely contributed to the variation we observed among trees.

## CONCLUSION

This study aimed to assess the influence of soil nutrient availability on sap flux density in northern hardwoods. Specifically, we sought to verify whether increased tree water use could explain the observed reduction in runoff following a calcium silicate application in an earlier study, and we tested for effects of N and P addition on sap flux density in a long-term N by P factorial addition experiment. Unfortunately, poor statistical power due to high tree-to-tree variability prevented detection of potentially important treatment effects. Previous studies that reported significant differences had larger effect sizes; few had sufficient statistical power to detect small differences. To improve the ability to detect a treatment effect, future studies should instrument more trees to increase statistical power, collect pretreatment data, when applicable, to control for tree-to-tree variability, and include additional tree metrics such as leaf and sapwood areas.

## ACKNOWLEDGEMENTS

Mark Green, Michele Pruyn, John Drake, and Pam Templer provided useful information on the mechanisms of sap flux and what variables to consider when monitoring tree water use. Matt Vadeboncoeur, Jamie Harrison, and Adam Wild helped in the design of this study by explaining previous design flaws and their own recent findings. High school and undergraduate students and summer interns helped build probes and instrument trees. Russell Briggs and Steve Stehman provided statistical advice. The two anonymous reviewers provided critical feedback for improving this publication.

### Funding

The MELNHE project was funded by the USDA National Institute of Food and Agriculture (2019-67019-29464) and by the National Science Foundation (DEB-0949324) including via the Long-Term Ecological Research Network (DEB-1114804 and DEB-1637685). The funders had no role in study design, data collection and analysis, decision to publish, or preparation of the manuscript.

### Grant Disclosures

The following grant information was disclosed by the authors:
USDA National Institute of Food and Agriculture: 2019-67019-29464.
National Science Foundation: DEB-0949324.
Long-Term Ecological Research Network: DEB-1114804, DEB-1637685.

### Competing Interests

The authors declare there are no competing interests.

### Author Contributions

- Alexandrea M. Rice performed the experiments, analyzed the data, prepared figures and/or tables, authored or reviewed drafts of the article, and approved the final draft.
- Mariann T. Garrison-Johnston conceived and designed the experiments, authored or reviewed drafts of the article, and approved the final draft.
- Arianna J. Libenson performed the experiments, analyzed the data, authored or reviewed drafts of the article, and approved the final draft.
- Ruth D. Yanai conceived and designed the experiments, authored or reviewed drafts of the article, and approved the final draft.

### Field Study Permissions

The following information was supplied relating to field study approvals (*i.e.*, approving body and any reference numbers):

The United States Forest Service gave us permission to conduct this experiment in the Jeffers Brook Watershed.

## Data Availability

The raw data is available at: Rice AM, Johnston MT, Libenson AJ, Yanai RD. 2021. Temperature differences in hardwood trees using thermal dissipation probes in Hubbard Brook Experimental Forest, NH and Bartlett Experimental Forest, NH ver 1. Environmental Data Initiative. https://doi.org/10.6073/pasta/a5d384c93682910d7246bb5bcbbcae9a.

## Supplemental Information

Supplemental information for this article can be found online at http://dx.doi.org/10.7717/peerj.14410#supplemental-information.

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
