# Peer review of "Tree variability limits the detection of nutrient treatment effects on sap flux density in a northern hardwood forest"

_PeerJ, doi:10.7717/peerj.14410_

## Round 0.1 · original submission · Major Revisions

Your manuscript has now been reviewed by two experts in sap flow measurements. Both found value and interest in your work but have identified issues that need to be addressed. Once you have addressed them, the manuscript will need to be reassessed before a new decision can be made.

In addition to their points, I would like you to clarify something about the data analyses. If I understand correctly, you used daily values of sap flow. Hence, several data points are likely to come from the same trees. If this is right, then this violates the assumption of independence between the observations. This can be addressed with repeated measurement ANOVA or mixed-effect model with the individual tree as a random term.

L180: beware that the changes in day length will impact your results

L242 sites and species are somewhat confounded. I don’t think it make sense to compare them as you cannot interpret the results (whether the difference come from sites, conditions as you mentioned or species).

Reviewer 1 ·

Basic reporting

Basic reporting meets journal standards; however, see comments to author for several suggested literature citations to add.

Experimental design

As the authors conclude, tree-to-tree variability was too large to detect treatment effects of fertilization, thus indicating that sample size was too small. I don't think that this is necessarily a flaw in experimental design, but given the limits on detection limits with their data, I would suggest that the analysis be expanded to examine other possible effects of fertilization on water use (see additional explanation in comments to author section).

Validity of the findings

no comment

Additional comments

Review for “Tree variability limits the detection of nutrient treatment effects on sap flux density in a northern hardwood forest” by Rice et al. (PeerJ 58823).
Rice et al. used sap flow sensors to test for fertilization effects on transpiration in several deciduous, hardwood tree species. This study makes use of a multi-year, full factorial experimental in three forests in the northeastern United States. They found no significant differences in sap flow among treatments; however, they identify that high variability among trees within a given group (species and treatment) overshadowed any potential changes.
The authors highlight an important challenge in estimating transpiration using sap flow probes (see for example Kumagai et al. 2005 suggested n=6; I suggest adding this citation). This problem is compounded in studies with multiple species and multiple experimental treatments, where the number of trees needed for appropriate sampling can be very high. I would agree with their conclusion that given the high tree-to-tree variability in sap flow, undersampling prevented a conclusive assessment of fertilization effects.
Overall, the paper is well written, using clear and concise language and logical flow. The figures and tables are informative and the captions provide useful explanations. Statistical analysis is well done and appropriate for the objectives and hypotheses.
My primary concern with this manuscript is that the results are inconclusive due to what is essentially a flaw in sample size. I do appreciate the authors’ approach in addressing this in a straightforward and quantitative manner. However, I would prefer to see some additional analysis of fertilization effects.
For example, in addition to comparing total daily water use, you could also examine if fertilization affects how water is used. As noted by Flo et al. (2019), thermal dissipation probes have relatively low accuracy, but are very useful in assessing relative sap flow. Thus, although these probes may not provide a good estimate of absolute sap flow (or flux density), they can provide a very good method for examining relative change in sap flow over time. So, it would be possible to test if there is a fertilization effect on diurnal water use trends or sensitivity to environmental drivers (e.g., differences when soil water may be limiting or when VPD is very high). These analyses could be performed on a relative or normalized scale and would be independent of tree-to-tree variability.
Other suggestions:
I’d encourage the authors to include reference to Ward et al., 2018 who used a large array of thermal dissipation probes and observed a modest decreased transpiration following nitrogen fertilization in loblolly pine.
Lines 69-
Lines 149-150: A few points on this sentence: (1) Some additional details are needed about your filtering criteria. Is there a quantitative threshold for “high photosynthetically active radiation”? And is the VPD threshold a peak daytime value or average daily value > 1 kPa? (2) I think this description needs to go in the “data processing” section. Presumably, data were collected on other days, as well—these are just the days where you have good data, right? (3) In general, focusing the comparisons on just days with ideal environmental conditions makes sense, but it would be useful to demonstrate that within-tree variability is low. This would confirm that you’ve selected a useful envelope of environmental conditions and that environmental variability isn’t contributing to the high CVs. (If I have misinterpreted your approach and you also included low PAR and low VPD days in the analysis, then that could present a problem.)
Lines 170-172: Did you consider potential nocturnal sap flow (transpiration or recharge)? As Lu et al. 2004 and subsequent papers point out, it may not be appropriate to assume that the maximum temperature difference each night is a “baseline point” (i.e., zero flow).

Minor comments:
Lines 41-45: May want to consider effects of nutrients on root production and how that might affect water uptake, especially during drought.
Line 200 and 205: It sounds like you are saying that “stand” was both a fixed and random effect. It seems like it should be a random effect for the purpose of this analysis.
Line 305-306: Do you mean to say that “the effect of calcium silicate addition on water use in a Hubbard Brook watershed was only a 25% reduction in streamflow”? And, if so, I would consider that a very large change.


References:
Flo V, Martinez-Vilalta J, Steppe K, Schuldt B, Poyatos R. 2019. A synthesis of bias and uncertainty in sap flow methods. Agricultural and Forest Meteorology 271 (2019) 362–374. https://doi.org/10.1016/j.agrformet.2019.03.012
Kumagai, T., Aoki, S., Nagasawa, H., Mabuchi, T., Kubota, K., Inoue, S., Utsumi, Y., Otsuki, K., 2005. Effects of tree-to-tree and radial variations on sap flow estimates of transpiration in Japanese cedar. Agricultural and Forest Meteorology 135 (1–4), 110–116. https://doi.org/10.1016/j.agrformet.2005.11.007
Ward EJ, Oren R, Seok Kim H, et al. Evapotranspiration and water yield of a pine‐broadleaf forest are not altered by long‐term atmospheric [CO2] enrichment under native or enhanced soil fertility. Glob Change Biol. 2018;00:1–16. https://doi.org/10.1111/gcb.14363

Reviewer 2 ·

Basic reporting

The introduction is clear and the point. The thing that is lacking is the importance of time and the uncertainty of the method for producing absolute values (Peters et al. 2018; https://nph.onlinelibrary.wiley.com/doi/full/10.1111/nph.15241). First, it could be that 3 years is not long enough for a mature forest. These type of legacies should be discussed and considered within the introduction (Zweifel & Sterck 2018; https://www.frontiersin.org/articles/10.3389/ffgc.2018.00009/full). I could image that 3 years is already quite extensive for a fertilization project, so one could use that as a strength of the study, although we should not have the illusion of replicating the decade long increased deposition of N. Second, I feel that way that the story is framed currently is to prove the impact of nutrient additional on absolute sap flow. Yet within literature it is widely discussed that absolute values obtained from sap flow are not fully valid, hence Dietrich et al. 2018 (https://besjournals.onlinelibrary.wiley.com/doi/10.1111/1365-2745.13051) refrains from using absolute values. This has not been discussed nor introduced. This is a large limitation of point measurements performed with sap flow probes and cannot be overlooked. Particularly in the context of the wide array of calibration curves recently published (Flo et al. 2019; https://www.sciencedirect.com/science/article/abs/pii/S0168192319301248?via%3Dihub) which significantly impacts the absolute values (Peters et al. 2020; https://besjournals.onlinelibrary.wiley.com/doi/full/10.1111/2041-210X.13524).

Experimental design

As state in my prior comment, the strength of thermal dissipation probes is not fully related to generating absolute values, but rather to explain sap flux density dynamics against environmental variables (Pappas et al. 2018; https://academic.oup.com/treephys/article/38/7/953/4993739). The response of sap flow to vapour pressure deficit or other relevant environmental parameters has not been assessed within the experimental design. When looking at the figures, it does appear that on a daily basis the nutrient addition might cause higher transpiration rates before noon. Such dynamics should be assessed as these could provide additional key insight on stomatal regulations and the adjustment of it. Moreover, to much emphasis is put within the manuscript on the issues in relation to the sensors. This is not needed. One can simply state that appropriate data was obtained for X days and sensors. Detailed reasoning as to why sensors did not work is not relevant for a physiological story presented here. Finally, I would argue that to little information is provided on how the thermal dissipation probe data is processed. This is key as it can heavily impact the daily sap flux density patterns (as illustrated by Peters et al. 2018).

Validity of the findings

Although I fully agree and appreciate the conclusion that absolute values vary to much between trees, I found little discussion on explaining what we should do in the future to reduce this variability. Additionally, as no contextual information is provided about the crown exposure and the stature of the trees it is really difficult to find reasons for this high variability. It is indeed really powerful to compare the results to literature values, yet one has to provide discussion as to why no patterns are found. Finally, I feel that in general the discussion is more a repetition of the results instead of putting them into the literature context. Moreover, although hypotheses are presented within the introduction, they are not assessed in depth within the discussion. I understand that not finding absolute effects does not allow to fully answer the hypothesis, yet one would then expect to find reasons as to why this is the case.

Additional comments

Line 44: This also depends heavily on legacy effects, which are nicely illustrated by Zweifel & Sterck 2018 (https://www.frontiersin.org/articles/10.3389/ffgc.2018.00009/full). Such legacies should be mentioned.
Line 66: These are indeed important considerations. Potentially the authors could add some information on what area on a global scale constitutes such N-limited conditions for forest ecosystems? Here I would expect the reader to become aware of the importance of these types of limitations.
Line 75: I highly appreciate the authors having concrete hypotheses which they are testing within this work. It would however be good for the reader if you stipulate what type of trees these are. Are these mature trees? Or juveniles?
Line 89: Are these natural forests or plantations? Please specify this from the beginning.
Line 93: Consider expressing the precipitation in mm. This is commonly done in literature.
Line 94: Well drained compared to what? Here I would either specify the soil water content range (or soil water potential) or not describe this. I assume the authors mention this as the soils are not water-logged.
Line 97: Here I miss a general description of the overall tree height and diameter and breast height. Additionally, I would be good to mention the overall stand density and understory growth.
Line 120-130: The principles behind Granier’s thermal dissipation probes have been described in a multitude of studies, thus I would not find this highly relevant to repeat. Yet when moving to the data processing, almost no detailed information is provided. It is unclear how the maximum DeltaTmax was established, if the users had to provide corrections, or if thermal drifts were accounted for (see Peters et al. 2018; https://nph.onlinelibrary.wiley.com/doi/full/10.1111/nph.15241). As such this will be difficult to fully be reproduced. Please clarify these details and put less emphasis on the method itself. I am aware the Baseliner is mentioned, yet this also required specific parameter choices.
Line 131: How was the health established? Looking at full crowns? Please specify.
Line : The number of tree monitored overall is quite impressive. One could think about highlighting this.
Line 119: I do not think the average statistics is relevant. I think one can just provide the range per plot and treatment.
Line 139: Why did the authors only consider monitoring for this short period of time? If there is a logical reasoning for this, please specify.
Line 150: I think the author mean that they solely used data from days with these conditions.
Line 156-163: Such details are not fully relevant. Additionally, one can correct for the sapwood/heartwood boundary (Clearwater et al. 1999; https://academic.oup.com/treephys/article/19/10/681/1632587) so I do not consider this a valid reason to exclude data.
Line 175-177: I would be really careful with such statements. I did see difference between diffuse porous species and I do not think that Bush et al. 2010 tests a large variety of different diffuse-porous species either. I would thus just state that you use a diffuse-porous specific calibration and assume species specific differences are minimal. This is a valid assumption and should be described as an assumption.
Line 179-186: Again this relates to issue which in the end just reduce the sample size. I would not go into too much depth on this.
Figure 1. A legend is missing explaining the different colours. These individuals, yet are not labelled. Moreover, when presenting the data one could also add the other treatments to provide an overview.
Line 231: One needs to provide statistics on this matter. Would it be possible to generate a table showing the statistical test results.
Line 254: These results all focus on the absolute amounts of sap flux density. Yet, the upscaling of sap flow measured with the probes is exposed to multiple issues and uncertainties (Lu et al. 2004; https://www.ishs.org/sites/default/files/documents/lu2004.pdf). Moreover, canopy exposition could hamper a proper comparison. As such, I rather feel that the strength lies in analysing the daily patterns. One could consider looking at the sap flow response to VPD and their differences. Has this been considered?
Line 270-277: Although I highly appreciate this use of literature, I feel like the authors should emphasize that for the addition of 3 years of fertilizer, these results are valid. One does not know about the longer-term effects, not about the delay at which the fertilizer is actually usable for the trees.
Line 279: Now the question becomes what other factors contribute the maximum amount differences. I still feel however that on should also explore the response difference between the treatments to vapour pressure deficit. Sap flux density is not verry well known for producing reliable absolute amounts. Also, when viewing Figure 2 it is clear that the NP trees reach higher maximum sap flow then the control. Has this been explored? This could be related to the lower sensitivity to vapour pressure deficit.
Line 312: One is able to approach stomatal regulation with sap flow data (Meinzer et al. 2013; https://academic.oup.com/treephys/article/33/4/345/1716840). Why has this not been tested with this dataset?
Line 330: No discussion has been made on the fact that mature trees might need a longer exposure, nor whether the added fertilization was sufficient to cause a difference. Additionally, no discussion has been made on other factors hampering the interpretation of absolute sap flux density data, like competition, leaf area to sap wood area. Such discussion points have to be included.

---

## Round 0.2 · Major Revisions

Dear authors,

Thanks for submitting a revision of your manuscript. I have now received reviews from the two original reviewers. Both reviewers are positive that the manuscript has been improved, but are also suggesting further improvements. I strongly encourage you to address their suggestions and comments since they are meant to improve the manuscript. I am looking forward to reading the revised manuscript.

Best regards

Yann Salmon

Reviewer 1 ·

Basic reporting

Manuscript is written in clear language and components (references, methods, figures) are well done.

Figure 6 is somewhat confusing. I did not see an indication that Bartlett-Mature was measured in 2018 in tables 2 & 3. Similarly, I could not determine if these data are all from one species or from different. If they are all one species, please specify in the caption. If not, please use different color lines to indicate data from which species are being shown.

Experimental design

Motivation for work is well articulated and significant. Experimental design is explained well and, despite noted data limitations due to high variability among sap flux sensors and other causes for data limitations, is appropriate. Limitations due to experimental design are clearly explained and accounted for in the analysis and discussion, so this section meets standards.

Validity of the findings

The findings are conclusions are appropriate, based on the data. Despite a general finding of no significant effects of treatments, I think publishing these types of studies are useful to (1) explain that, contrary to the hypotheses, there may not be strong evidence of a treatment effect and (2) to highlight limitations with data and provide suggestions for future work. Specifically, physiological variability and technical challenges can lead to high variability among sap flux measurements. This is particularly true in forests with diverse species and a variety of treatments, where a very large number of sensors are required to capture statistical significance.

Please see "additional comments" for some further suggestions.

Additional comments

This is my second review of the manuscript by Rice et al. and while I found the authors made significant improvements to the manuscript, I still have several questions and suggestions.

If I understand your ANOVA analysis and tables (5 & 6) correctly, it looks like you do not consider “species X treatment” interactions. Thus, you are (correctly) testing for effects of treatment on sap flow, as well as interspecific differences in sap flow, but are not evaluating whether different species respond differently to treatments (e.g., if red maple increases sap flux with N-fertilization, but white birch doesn’t). Similarly, I think you need to include interactions with “stand”, to test whether the response to treatments varied among sites (e.g., if sap flux increased in sugar maple with N-fertilization only at Hubbard Brook-Successional, but not at Bartlett-Mature). Thus, you should include treatment X site as well as treatment X species X site. This approach would be the most complete and can test for site-specific responses that may exist based on data shown in figures 2-4.

It is unfortunate that VPD was only measured at one site (Bartlett NEON), because this severely limits some analyses both I and the second reviewer suggest which could help to identify treatment responses in sap flow. It is not surprising that there is no significant sap flow response to VPD, given that you have filtered data to only include high PAR/high VPD days (i.e., sap flow should be fairly consistent among days with these conditions, absent any drought). I am not sure how VPD was used as a data filter if it wasn’t measured at other sites—perhaps it was a measurement from a nearby climate station that might not represent the local stand? In any event, I think that if data for days with lower PAR and VPD were included, it might be possible to test for species- and treatment-level sensitivities to environmental forcing variables. However, given the frequent sensor problems, it is unclear if there are enough data to pursue these types of analysis. The one recommendation I do have is that if soil moisture data are available for any of the sites (even at a “site-level” if not precisely co-located with the sap flux study plots), you can test for a high- versus low-soil moisture effect. This would help to address the rationale of understanding how drought, combined with fertilization, might affect plant water uptake. In other words, does fertilization lessen or strengthen a species’ sap flow response to drought (species X soil moisture X treatment interactions).

One set of issues that should be addressed are with the framing of the paper’s objectives: (1) Green et al. noted an initial increase in ET of 25%, which decreased to 18 and 19 % in the next two years, and they note that the peak of 25% may have been a transient response. So, perhaps <20% may be a more characteristic response for fertilization. However, these results were the initial responses in 2000-2002, and it is unclear whether these differences have persisted or changed over a decade and through your current experiment. Theoretically, fertilization effects on water use may have declined over time. I haven’t done an extensive study of HBEF streamflow, but surely recent ET estimates are available to add to this analysis. I suggest including more recent ET estimates to determine the magnitude of ET response you might actually see. (2) It is also worth considering that the negligible response of sap flow to fertilization could be correct and, if so, what mechanisms could lead to increased ET. Green et al. also note a strong increase in aboveground biomass and LAI. The increase in biomass, if it is in the corresponds with higher total sapwood area, could produce higher transpiration even with the same sap flow rates among treatments. For this aspect, you present basal area for the different stands—can you calculate sapwood area, based on species-specific allometric equations (assuming no change in allometry with treatment) to get a rough idea if differences in sapwood area alone could explain differences in streamflow and total ET. Similarly, the increase in LAI could also lead to higher interception, thereby increasing evaporation and ET independent of transpiration. So, rather than focus solely on the potential technical/methodological issues which may have obscured a fertilization response (which are important and should be left in), also expand the discussion to consider whether the magnitude of other changes to the system could affect ET.

Other comments
Line 60-61: This statement is not accurate. Ward et al. (2018) showed that N fertilization led to a significant, albeit small, decrease in transpiration and ET.

Lines 169-170: how were PAR and VPD measured? You mentioned in your response that VPD was only measured at one site. Include information on the location of these measurements in proximity to the plots.

Reviewer 2 ·

Basic reporting

The manuscript entitled: “Tree variability limits the detection of nutrient treatment affects on sap flux density in a Northern hardwood forest” addresses a timely topic in relation to mature forest responses to nutrient addition. This is the second time I reviewed the manuscript and considered both the response to the referee comments and the newly provided version of the manuscript. Although the authors have put substantial effort in addressing the comments, I feel that especially comments in relation to physiological explanations for the absence of an effect on absolute sap flux density have not been sufficiently addressed. Within the discussion the authors now mention some mechanisms which could impact absolute sap flux density, yet fail to discuss why these allometric properties (i.e., leaf area vs. sapwood area) or functional traits (i.e., stomatal conductance) would not change despite the nutrient addition, or potentially not impact the sap flux density. The main message is that the variability is high, yet this also shows that even after a long fertilization experiment that the treatment effect is not strong enough to overcome these limitations (although other studies have found differences). This is particularly apparent in the discussion which still mainly repeats the research findings without spending more time on the interpretation of the results. I would for instance have expected a more dedicate discussion on that fact that despite finding changes in the diameter (as debated in the introduction) this did not lead to increased sap flux density. One explanation could be that rather the sapwood area increased instead of the flux per unit sapwood area, as wood anatomical properties did not change and allow for stronger fluxes. Such discussions are not present and should be considered. Moreover, highlighting experimental design differences (besides statistical method differences which are now presented) between this study and the studies presented within the literature review (Table 7) which did find significant changes could point to future improvements that could be discussed.

Experimental design

Multiple comments were made on the fact that the sampling design is not fully sufficient for drawing conclusions on absolute sap flux density. Upon closer reading of the manuscript I in fact only now noticed that actually 3-7 day periods were extracted from different periods in time with likely contrasting environmental conditions. First, I would thus suggest the authors to make a clear graphical representation of the extend of the actual data used in the analyses (i.e., an appendix figure). Second, I could find no description of the difference in the environmental conditions from which the data originates in time (i.e., differences in environmental conditions of the selected days). Although the discussion mentioned that environmental variability could have an impact, no effort was made to actually quantify this. Finally, for the calibration curve uncertainty, all calibration curves are made available in Peters et al. (2021; Methods in Ecology & Evolution; https://besjournals.onlinelibrary.wiley.com/doi/full/10.1111/2041-210X.13524) and could potentially aid the authors in reducing uncertainty (if their species have been reported).

Validity of the findings

Within the manuscript the authors often make statements that they could not prove that there is no positive effect of nutrient addition on sap flux density. For example: “It remains possible that nutrient availability has an ecologically important effect on sap flux.”. Yet, within this study no significant effect was found and it now reads as if the sole explanation is effect size issues. It could however also be true that the effect in this particular forest or on these particular species is not uniformly strong. As such I am wondering why the authors still expect a positive response. Would it not be better to be more careful with such statements within the discussion and the abstract (i.e., “we cannot eliminate the possibility of small but important effects of nutrient availability on tree water use.”)?

Additional comments

All minor comments made here refer to the line number provided in the track change Word version.

Line 26-27: It is indeed good to mention within the abstract that specific stem and leaf properties have changes due to the nutrient addition. Here I do however wonder whether the total leaf area, or SLA, changed as this will likely have a stronger effect on the absolute amount of transpiration.

Line 41-43: Is this conclusion built upon the literature you analysed? Because the results presented within the manuscript actually show the opposite. Would it not be better to conclude that sample size in such fertilization experiments is of critical importance? Moreover, I would urge the authors to mention that their results also should motivate the sap flow community to more critically assess factors which could reduce uncertainty within absolute sap flow measurements.

Line 61: You mean poorly understood.

Line 69: Here I would have expected the authors to clarify what could cause these differences and how their experiment could aid in solving some of the uncertainties.

Line 97: For absolute sap flux density to increase, a tree likely has to adjust the wood anatomical sapwood properties which have not been mentioned here or linked to the nutrient treatment. Moreover, I am still quite surprized that no mention is made about potential changes in stomatal control (see for example Ewers et al. 2001, Tree Physiology: https://doi.org/10.1093/treephys/21.12-13.841).
Line 109: Missing space before Table 1.

Line 127-128: Please specify the temporal extend more clearly. It is not specified which months were considered for monitored. Also, it is unclear from the description how many days in total were considered.

Line 142-144: I do not agree with this statement. Calibration curves significantly differ between species, and particularly plant functional types (i.e., ring-porous vs. diffuse-porous species). The different calibration curves are provided by Peters et al. (2021, Methods in Ecology & Evolution; https://doi.org/10.1111/2041-210X.13524), which clearly shows differences in the steepness of the calibration curves. The study by Flo et al. (2019) reported a fixed offset which they used in a large-scale meta-analyses of the SAPFLUXNET as they lacked calibration curves for all species. I would thus suggest the authors to make more nuanced statements about the calibration impact.

Line 172-173: This is confusing to me. Above multiple years of monitoring are mentioned. Then here one reports on the fact that a maximum period of a week was used. This should be clear from the beginning. Also, please report on how many days in total are now considered, when considering the data gaps.

Line 187: K value calculations are normally presented as (ΔTmax-ΔT)/ΔT (see Ganier 1985). Also it is not clear whether zero-flow conditions were fixed for each 3-7 days periods or whether it could vary between days.

Line 195: The calibration curves are provided in the TREX R package (see: https://cran.r-project.org/web/packages/TREXr/index.html). Moreover, I don’t think Steppe & Lemeur (2007) is the correct citation here as they used Dynagage sap flow sensors (SGA5 and and SGB16) sensors. Moreover, this is advanced modelling study showing the importance of wood structure on flow resistances. They did not discuss species-specific calibration curves.

Line 204: So if I understand correctly daily average sap flux densities were extracted? While one could have also calculated the more commonly used daily sum of sap flow (see Wullschleger et al. 1998, Tree Physiology: https://doi.org/10.1093/treephys/18.8-9.499). Please clarify which parameter was used, as within Figure 2 the authors show the daily sum of sap flux density, not the average.

Line 218: Specify whether this is daily average sap flux density.

Line 274: Indeed the environmental conditions between the monitoring conditions could have been substantially different. From the experimental design it is also not really clear how different the environmental conditions were between the weeks that were effectively selected for the analyses. I think the manuscript would benefit from an additional appendix figure showing the variability in environmental conditions for the different monitoring periods. Moreover, it would be good to add a graph with the temporal extend for each tree and treatment. Making a bar graph with time on the x-axis and the trees and treatment on the y-axis could help. Then readers can identify the temporal overlap between measurements.

Line 310-311: This again seems to be something that belongs within the methods and not in the result section.

Line 320: Within the method section one should clarify how the literature search was performed. I presume this was a search with specific key words which could be mentioned.

Line 325: One needs to remove “;”.

Line 340-342: Within the introduction clear objectives are provided, which were addressed in this study. Yet, here a new reasoning is stated as to why the study was performed which should have been described in the introduction. Additionally, one would normally start with postulating the main finding of the work and its implications, not solely mentioning a single stand and a specific result found there.

Line 346-348: I still do not understand how one can both state that the effect size was too low to see any effect, while still arguing that you cannot rule out the possibility for a sap flow increase by specifically 25%. I would rather argue that the study is inconclusive and one can not argue for or against and increase in sap flux density.

Line 349-352: As stated in my previous review, the discussion here reads a lot like a repetition of the results and not like a proper discussion on the found patterns. Reporting on the strength of the P value is not the exciting part for the reader. How we can interpret the results is! Below the authors already state that the results are not highly convincing (Line 355), so should we then conclude that nutrient addition does not affect sap flux density? What physiological reasons could there be for not finding a strong increase in sap flux density?

Line 371-372: Why is the statistical method discussed in so much detail here. I understand one wants to consider the reasons as to why an effect is not measured. However, there is no discussion as to which physiological reasons there are for not seeing a stronger increase in sap flux density.

Line 387-388: So no effect was found within the results which is mainly attributed to the sample size. Yet, the alternative could also be true. That the effect is really small, or does not affect the absolute sap flux density as much as other factors, like stomatal control. Why do the authors now still conclude that it remains possible that nutrient addition is important for sap flux?

Line 394: Which figure?

Line 402: Within the methods it is clear that environmental data is available. Why has this variability in environmental conditions between monitoring periods not been quantified.

Line 423: This is a conclusion not a summary. Focus this part on the main findings.

---

## Round 0.3 · Major Revisions

Dear authors,

Your manuscript has been reviewed once more by an expert in sap flow measurements. This is a very detailed review (as were the earlier ones) which clearly aims at improving the manuscript. In my opinion, the issues raised by the reviewer (which were already mentioned in the previous reviews) are valid and important. I share the reviewer's concern that you cannot properly address the following: “One goal of this study was to determine the role of water use by trees in reducing run-off
for 3 years after treatment as postulated by Green et al. (2013)” and “the importance of other nutrients to tree water use” without addressing the methodological and logical limitations in your study either with further analyses as suggested earlier or at the very least by discussing them in depth. In both cases, the key issue is that you cannot directly extrapolate tree water use from sap flow density if you do not account for changes in conducting sapwood area.

Furthermore, I agree with the reviewer's statement that “the burden is on the authors to build support for their premise by working to eliminate other likely explanations that might produce a similar result” when making some conclusions. For example, you write that “Our second objective was to determine whether this failure to detect effects with greater statistical confidence was due to small effect sizes or high variability among trees” This is an important question, but there is another possible explanation: that there is actually no effect either because there is no response of the trees, or because the response is something you did not measure. Only when these cases have been addressed, can the question above be convincingly answered.


I urge you to thoroughly address the concerns raised in this review and look forward to a revised version of the manuscript.

Best wishes

Yann

Reviewer 1 ·

Excellent Review

This review has been rated excellent by staff (in the top 15% of reviews)
EDITOR COMMENT
The reviewer build a well reasoned and thorough case on how the authors could greatly improve their manuscript and reinforce the validity and scientific relevance of their findings. The reviewer discuss the issues of the manuscript in a detailed and constructive manner and provides the authors with clear guidance on what they have to do to address each of the issue identified by the reviewer. I would like to thank the reviewer for the patience, effort and expertise underlying this excellent review.

Basic reporting

Basic reporting is well done. Language is clear, manuscript is well structured, and figures are high-quality.

Experimental design

Overall experimental design is solid. Analysis and subsequent results would be stronger if authors had more days of sap flux data, spanning a wider range of environmental variability, along with related soil moisture data; however, these types of limitations are often unavoidable. I do believe that there are other relevant data available to frame the objectives and address the potential mechanisms. Specifically, can any longer-term streamflow data or publications from fertilized plots at Hubbard Brook be leveraged to explain whether the initial increase in ET observed by Green et al (2013) persisted, returned to pre-treatment levels, or even declined? And, can basal area, or preferably sapwood area, be incorporated to explain another key mechanism that may be affecting transpiration. (See additional explanation below.)

Validity of the findings

I am still concerned with what I consider logical and methodological flaws in this paper. The paper aims to test whether decreased streamflow following fertilization observed in a previous study was caused by increased transpiration. Increased transpiration could occur through a variety of mechanisms, including but not limited to: increased sapwood area, increased hydraulic conductance of sapwood, or temporal changes in transpiration. This paper uses sap flow probes (sap flux density measurements) to examine only the possibility of increased hydraulic conductance, largely ignoring other possibly mechanisms. However, I will note that because the analysis uses daily total sap flux, it actually combines instantaneous flow rates and diurnal temporal trends, obscuring the potential mechanism. In other words, increased daily sap flow could be due to higher peak sap flux density (i.e., higher hydraulic conductance) and/or transpiration occurring over more time throughout the day (e.g., due to greater conductance when other trees become hydraulically constrained, or through greater afternoon/evening recharge). In a previous review, I suggested examining these diurnal dynamics, but this has not been incorporated. Because the data for this study are limited to only high-PAR and high-VPD days, there is not a way to examine intra-annual temporal dynamics that might contribute to higher transpiration, such as the ability for trees within a certain treatment to maintain higher levels of transpiration during drought than other trees. This is unfortunate, but understandable—although should be noted in the discussion as another possible mechanism.

However, the greatest deficiency in this paper is the failure to examine the possibility that the increased basal area observed after fertilization (as you cite with Goswami et al., 2017 and Green et al., 2013) could explain some or all of the increase in transpiration. From a physiological perspective, increased basal area and sapwood area is a much more straightforward hypothesis and, given the extensive data from this study, should be easy to test. Even if there are not data available for analysis, the authors need to explain how fertilization is expected to affect sapwood area and transpiration in the Discussion section.

While the manuscript text does a good job of explaining that there are some weakly-significant results that suggest possible treatment effects on sap flux density, the authors’ rebuttal letter seemed quite adamant that the treatment effects are real and that no other potential mechanisms need be examined. I agree with the author’s comment to Reviewer 2 that “It is a common misconception that failing to detect a significant effect means that there was no effect”; however, the logical extension of this statement is not conclude that there was an effect. Instead, the burden is on the authors to build support for their premise by working to eliminate other likely explanations that might produce a similar result. (I’ll add that even if the authors found a highly-significant treatment effect on sap flux density, that they should still consider other mechanisms, especially changes in sapwood area, which might enhance or counteract effects on sap flux density.)

I would also like to revisit the results from Green et al. (2013) and how it relates to this work. As you mention in the abstract, Green found a 25% increase in ET the first year following fertilization; however, this response showed a slight decline for the next two years 18% and 19%) after which ET returned to pre-treatment levels. Thus, because the measurements in this study take place several years following fertilization, an equally valid hypothesis could be that transpiration may not differ among treatments and that sap flux density may actually be lower in fertilized plots to compensate for higher sapwood area. I haven’t reviewed the literature enough to know what the longer-term effects of fertilization on streamflow was for these sites, but any information pertaining to that would be relevant. (And I do understand that the sites used in Green’s study are different than yours.)

The analysis assumes that treatment effects on sap flux will be uniform among all species and all sites. However, I don’t think there is a theoretical basis for this assumption. Additionally, hypothetically, one species could demonstrate a strong increase with fertilization, another could show a small decrease, and others may show no change; there would likely be no statistically significant net effect of fertilization on sap flux, but this could lead to a change in transpiration at the stand level. The ANOVA, as presented in the methods and Table 5, tests for the effects of (1) time since treatment, (2) treatment (calcium silicate), (3) time since treatment X treatment, and (4) species on sap flux (site is a random variable). Similarly, Table 6 test for effects of (1) treatment, including NXP interactions, (2) species, and (3) stand on sap flux. Figure 2 appears to contrast this assumption: calcium silicate addition appears to increase sap flux in red maple in 2018_Hubbard Brook Successional stand, whereas many other species/stand combinations may be non-significant. Therefore, I would argue that you need a species-treatment interaction term.

Similarly, I am not convinced that Site should be a random variable. These aren’t replicate plots within a common area, but distinctly different forests with different local climate, soil properties, site histories, etc. Thus, we could hypothesize that sap flow differs among sites based on unobserved factors and, subsequently, that interactions among site, species, and fertilization may also affect sap flow in non-uniform ways. Based on my reading of the Goswami (2017) paper, there were plot-level differences in basal area response to treatments. Therefore, I would argue you also need a site-treatment interaction term, as well as a site-treatment-species interaction term. Even if these interactions are non-significant, they are worth reporting to demonstrate the full model results.

Additional comments

Line 162-163: The approach using Baseliner software you described will adjust for thermal drift, even over longer periods of time, so this statement is not necessary.

Line 176-177: “they” refers to what in the statement “if they did not follow the characteristic diurnal curve”? Is “they” the sap flux trend, PAR trend, both?

---

## Round 0.4 · Minor Revisions

Dear authors,

Your last revision addresses the major concerns that were raised earlier, and only very minor changes are required before I can accept the manuscript for publication.
I agree with the reviewer that the analyses presented in the rebuttal letter should be made available to the readers since they provide relevant information. In particular I agree with the reviewer that statistical test should be started from the full model (including interactions) and that reporting the non-significant interaction is relevant to your study. This could be done in just a few sentences.
I am looking forward reading the final version of the manuscript.

Best wishes

Yann

Reviewer 1 ·

Basic reporting

I have reviewed several previous versions of this manuscript and have concluded Basic Reporting is well done and satisfactory.

Experimental design

As previously noted, experimental design is satisfactory.

Validity of the findings

The authors have addressed my previous concerns and the validity of findings is now satisfactory.

Additional comments

There are two minor revisions that I would like to see the authors make to the manuscript. In the rebuttal letter, they included results from several suggested analyses. The results did not show statistical significance and were not included in the text of the main paper. However, I believe that including these analyses were important steps and their results should be mentioned. This should only take up a few sentences similar to what was provided in the rebuttal and, as mentioned, the analyses have already been performed by the authors. Specifically:
(1) For diurnal trends in sap flux, you explain "we tested if there were treatment effects at each hour of the day". Please make mention of this in your manuscript to demonstrate to readers that treatments did not affect diurnal water uptake dynamics. (Although you claim that this results is "disappointing", it actually helps to support the thoroughness of your approach.)
(2) I explained the need to incorporate interaction effects into the ANOVAs. You note that the effects were not significant; however, I believe running regressions with the full model (including necessary interaction effects) is the correct approach and reporting these non-significant effects are important. Please make mention of these results (data from the tables in the rebuttal letter could be incorporated into the text or tables could go into supplemental information sections). I have no problem with you then explaining your reduced model with better AIC values.

---

## Round 0.5 · accepted · Accept

Dear authors,

You have now addressed the reviewers' comments and the manuscript is suitable for publication. It would be fair to add the two anonymous reviewers in the acknowledgment since they put a lot of effort into helping you improve the manuscript. Thanks